# Geochemical Composition of the Lomé Lagoon Sediments, Togo: Seasonal and Spatial Variations of Major, Trace and Rare Earth Element Concentrations

**Tchaa Esso-Essinam Badassan** [1,2,*]**, Akouvi Massan Duanyawo Avumadi** [1,2]**,**
**Kamilou Ouro-Sama** [2] **, Kissao Gnandi** [2]**, Séverine Jean-Dupuy** [1] **and Jean-Luc Probst** [1,*]

[1]  Laboratory of Functional Ecology and Environment, University of Toulouse, CNRS, UPS, Toulouse INP, Campus ENSAT, Avenue de l'Agrobiopole, 31326 Castanet Tolosan, France; havumadi@gmail.com (A.M.D.A.); severine.jean@ensat.fr (S.J.-D.)
[2]  Laboratory Gestion Traitement et Valorisation des Déchets (GTVD), Faculty of Science, University of Lomé, Lomé B.P 1515, Togo; ouro_kamilou@yahoo.fr (K.O.-S.); kgnandi@yahoo.fr (K.G.)
[*]  Correspondence: badassan13@gmail.com (T.E.-E.B.); jean-luc.probst@toulouse-inp.fr (J.-L.P.)

**Abstract:** The concentrations of major, trace (TE), and rare earth (REE) elements and their seasonal and spatial distribution were studied on the fine fraction (<63 μm) of the sediments of the Lomé lagoons (West Lake, East Lake, and Lake Bè). The sediments were collected on a total of nine sampling sites (three per Lake) during two campaigns (dry season and rainy season). The quality of the sediments was assessed on the basis of the enrichment factor (EF) and the labile or non-residual fraction (NRF) in relation to the values recommended for the quality of the sediments (Sediment Quality Guidelines, SQG). The distribution of rare earth elements shows enrichments in light rare earths superior to those of heavy rare earth elements during any season. Positive Ce anomalies are less noticeable and less variable between seasons than Eu anomalies. La/Yb ratios are positively correlated with the percentage of Al and Fe oxides and with the percentage of fine fractions. The main bearing phases of rare earth elements are, therefore, Al and Fe oxides and the finest fractions of the sediments. The concentrations of trace elements vary little, according to the seasons, but show strong variations from one element to another. The degrees of enrichment obtained are moderate for Bi, Cr, Ga, Mo, Pb, Sn, and Zn (1.5 < EF < 5) to significant for As, Cd, and Sb (5 < EF < 20) for all sites of Lake Bè. For the sites of West Lake, the degrees of enrichment obtained are moderate for As, Cd, Cu, Mo, and Pb (1.5 < EF < 5) to a significance for As, Bi, Cd, Pb, Sb, Sn, and Zn (5 < EF < 20). Only the East Lake sites show high degrees of enrichment for elements such as Sb and Sn (20 < EF < 40). Trace elements (TE) such as As, Cd, Cu, and Ni have total concentrations within the range of variation of the SQG concentrations (particularly Probable Effect Level (PEL) and Effect Range Median (ERM)), whereas Cr, Pb, and Zn total concentrations are higher. The ranking of priority sites with respect to the sediment contamination is determined according to ERM and PEL quotients in relation to the probability of toxicity for benthic organisms. For almost all the sites, the priority is lowest to medium-low with regard to As, Cd, and Cu and medium-high (Cr and Ni) to highest (Pb and Zn), particularly for the East and West Lakes. Moreover, the NRF can represent significant percentages of the total TE concentrations: 5% to 15% for As, Bi, Ni, V, Mo, and Sc, 15% to 25% for Co, Cu, and Sr, 25% to 40% for Pb and Zn and, lastly, 47% to 55% for Cd.

**Keywords:** bottom sediments; trace elements; rare earth elements; seasonal variations; labile fractions; enrichment factor; sediment quality guidelines; brackish waters; lagoon; urban environment; West Africa

## 1. Introduction

Natural processes, such as rock weathering and soil erosion and anthropogenic perturbations including urban, industrial, agricultural activities, discharge of untreated wastewater, and leachate from landfills are the main sources of sediment contamination by trace elements [1–4]. Whatever their origins, trace elements, in addition to major and rare earth elements, are transported by runoff in dissolved or particulate forms and finish their transfer pathway by deposition on the bottom sediments in aquatic ecosystems [5,6]. The sediments then serve as a sink for potentially toxic elements [7,8] and store them through complex processes of physical and chemical adsorption [9,10]. Under these conditions, they can limit bioavailability or allow the remobilization and re-suspension of trace elements in the water column [4,10,11]. Once released into the water column, they represent a threat to aquatic ecosystems and a health risk to millions of people due to their toxicity as well as persistence and ability to accumulate in living organisms [12–17]. Thus, the characterization of sediment composition in terms of major elements, traces, and rare earth elements is useful and provides basic information on the contamination degree and ecotoxicological risk [18,19] for better management of aquatic ecosystems. Coastal lagoons are among the most used ecosystems [20] and, therefore, the most fragile, vulnerable, and threatened [21–23]. The Lomé lagoons in Togo are subject to high anthropogenic pressure, which is often at the origin of various organic and inorganic pollutant discharges, which contributes to the contamination of the lagoon ecosystem. Multidisciplinary studies in the past have made it possible to carry out a geochemical characterization of sediments [24–26] without being interested in the temporal variations of major, trace, and rare earth element concentrations in the sediments of the Lomé lagoon. Taking into account seasonal variations in the study of the quality of aquatic ecosystems is essential and has been the subject of several studies [5,27–31]. The particularity of this study is based on the integration of temporal and spatial variabilities in the distribution of major, trace, and rare earth elements in the sediments of the Lomé lagoon. This study also has the advantage of having a database not only updated, but robust on the state of pollution of the Lomé lagoon. The main objectives are:

- to characterize the micro-granulometry of the sediments to assess the particle size distribution,
- to evaluate the spatio-temporal variability of major, trace, and rare earth element concentrations,
- to determine the Ce and Eu anomalies and La/Yb ratios to evaluate the Rare Earth Element (REE) fractionation and its controlling factors,
- to assess the sediment quality using enrichment factors calculated using different normalizing elements (Al, Cs, Fe, Ti) and a local reference material,
- to assess the potential ecotoxicological risks for aquatic ecosystems by comparing TE concentrations with Sediment Quality Guidelines (SQG) and by determining the percentage of a non-residual fraction or labile fraction of TE.

## 2. Materials and Methods

### 2.1. Study Site

The Lomé lagoon system is located in the extreme southwest of Togo, bordering the Gulf of Guinea (Figure 1 and Table 1). It is bordered in the West by the Ghana border. Administratively, it is located in the maritime region and precisely in the town of Lomé. From east to west, the lagoon system of the city of Lomé consists of three natural lakes: Lake Bè, with a total area of 29 ha, Lake East (31 ha), and Lake West (20 ha). Lake East and Lake West are connected to each other by an equilibrium channel that extends over 1.2 km [25,32]. The three lakes are connected to the sea by small channels constructed to drain the lakes into the ocean during the high-water levels, but during the low waters, there are some ocean water intrusions into the lakes. Then the lakes are not fresh water but brackish waters with total dissolved solids varying between 2.1 and 2.8 g·L$^{-1}$ according the season.

The study site enjoys a sub-equatorial climate characterized by four alternating seasons of unequal length. There are two dry seasons (November to March and August) and two rainy seasons (April to the end of July and September to mid-November) [33].

The ferralitic soils in the study area are characterized by the predominance of dissolution, hydrolysis, and oxidation processes. These processes completely transform the parent rock and give rise to a clay, which is a kaolinite with a high proportion of iron and aluminum hydroxides.

**Table 1.** Lakes studied and geographic coordinates of the sampling sites.

| Lakes | Sampling Sites | Geographical Coordinates (UTM) | Altitude (m) |
|---|---|---|---|
| | S4 | X = 1.25543, Y = 6.15295 | 4 |
| Lake Bè | S5 | X = 1.25173, Y = 6.15187 | 4 |
| | S6 | X = 1.24868, Y = 6.15073 | 4 |
| | S7 | X = 1.24283, Y = 6.14925 | 2 |
| East Lake | S8 | X = 1.23909, Y = 6.14805 | 2 |
| | S9 | X = 1.23597, Y = 6.14692 | 2 |
| | S10 | X = 1.20060, Y = 6.13489 | 2 |
| West Lake | S11 | X = 1.20609, Y = 6.13668 | 2 |
| | S12 | X = 1.20963, Y = 6.13133 | 2 |

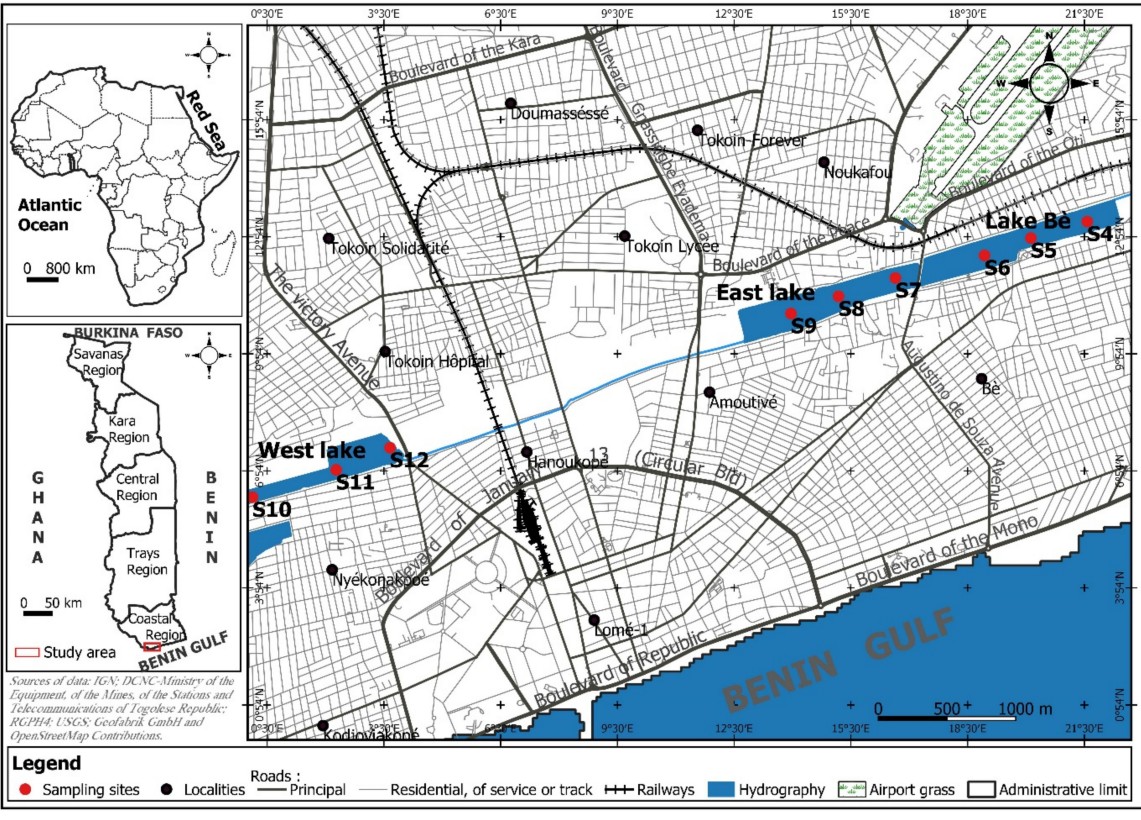

**Figure 1.** Geographic location of the Lakes and sampling sites in the study area of Lomé urban agglomeration.

## 2.2. Sampling and Processing

To achieve our objectives, the sediment samples were taken in two separate campaigns. The first campaign in the rainy season (July 2016) and a second campaign in the dry season (March 2017). In the field, the samples were collected using a small plastic grab sampler in the first centimeters of the sediment layer in accessible places and present a certain amount of fine deposits covered by the aqueous phase. The sediment samples were placed directly in plastic bags, labeled, and transported

rapidly to the Waste Management and Recovery Laboratory (GTVD) of the University of Lomé for treatment. Then, the sediments were dried in a Memmert electric oven type UL80 at 40 °C for four days in the Geology Laboratory of the University of Lomé. Once dried, the sediments were packed in plastic bags, labeled and transported in France to the Laboratory of Functional Ecology and Environment where the samples were finely crushed using a mortar, and an agate pestle in order to disaggregate the sediments. Between each sample, the pestle and mortar are cleaned and washed with milliQ water to avoid any contamination. Then, the sediments were sieved through nylon sieves of variable mesh (2 mm, 200 μm, and 63 μm) in order to separate the different particle size fractions of the sediments (clays and silts < 63 μm, 63 μm < fine sands < 200 μm, 200 μm < coarse sands < 2 mm and gravels > 2 mm). To perform the particle size analysis of each fraction, the Horiba LA 950 laser microgranulometer (HORIBA Company, Kyoto, Japan) from the Laboratory of Functional Ecology and Environment was used. Before reading, each fine portion of a sample is mixed with a solution of sodium hexa-meta-phosphate to improve the dispersion of particles in the solution.

The rest of the chemical analyzes (major, trace, and rare earth elements) were carried out only on fractions smaller than 63 μm. This overcomes the particle size influence in data processing. This standardization also finds its advantage because the finest fractions of sediments have large specific surfaces, and, therefore, a greater adsorption capacity than coarse sediments [34–36].

## 2.3. Analytical Methods

The determination of the total concentrations of major, trace, and rare earth elements in the sediments of the Lomé lagoon was carried out by the National Service of Rocks and Minerals Analysis (SARM: http://helium.crpg.cnrs-nancy.fr/SARM/) in Nancy, using the ICP-OES iCap 6500 and the ICP-MS iCapQ (Thermo Fisher Scientific, Waltham, USA). The sediment digestion technique used (alkaline fusion) consists of the fusion of at least 200 mg of the sample with lithium metaborate (LiBO$_2$) before analysis by the ICP. Furthermore, to validate the results of the chemical analyzes, international reference materials (WQB-1, SRM 1646–2) were used by following the same mineralization protocol as the sediment samples. Uncertainty becomes important over a concentration range located between the detection limit and the lowest concentration for which a percentage of error is indicated. In the case of this study, the uncertainty on all the measures is <20%. This made it possible to validate both the protocol and the results obtained during this study.

In order to determine non-residual fractions (NRF, labile phases) adsorbed onto the sediment, a simple extraction with EDTA (Ethylene Diamine Tetraacetic Acid) was used. The procedure consists of leaching 1 g of the fraction <63 μm of sediment with 10 mL of an EDTA solution of 0.05 mol per L$^{-1}$ at room temperature with stirring for one hour. The mixture obtained was filtered using 0.22 μm pore-size membrane filters. The filtrate is then analysed by the ICP-MS Agilent 7500ce (Agilent Technologies 7500ce, Les Ulis, France) on the analytical platform of the Midi-Pyrenees Observatory (OMP) in Toulouse.

This method has proven effective for evaluating non-residual fractions in several studies [37–42] because of its chelating power and its ability to extract trace elements from non-residual fractions. To validate this time, the chemical analyses of the EDTA extraction, which is an international reference material (SRL6), was analysed under the same conditions as our solution samples. The yield obtained between the measured and certified values of the reference material is very accurate. The recovery values range from 63% to 119% for samples collected in the dry season and from 62% to 103% in the rainy season. Errors on measures calculated for 20 elements were low (<4.10$^{-1}$).

## 2.4. Data Processing

### 2.4.1. Enrichment Factor

Since the 1970s, the enrichment factor (EF) has been commonly used to highlight possible contamination in certain environments. Its calculation is used to evaluate the degree of anthropogenic

pollution by trace elements [43]. Since then, this concept has been used by several authors [10,26,42, 44–48], whether on soil or sediment to determine the anthropogenic contribution in terms of trace element pollution. However, concerns exist regarding the choice of the reference element (normalizer) and the choice of the reference material (uncontaminated) because, according to References [49,50], these two choices determine the reliability and representativeness of the enrichment calculated in the regional context. The enrichment factor (EF) is calculated according to the formula of Chester and Stoner [43].

$$EF = ([X]/[R])_{sample}/([X]/[R])_{reference\ material}$$

X = concentration of the trace element studied

R = concentration of the normalizing element

Thus, the PAAS (Post Archean Australian Shale, [51]) and the UCC (Upper Continental Crust, [52]) are the reference materials frequently used and recognized as such by the scientific community [26,40–42,50,53,54] and easily accessible in the published literature. However, the use of these reference materials can lead to overestimations or underestimations of EF [45,50,55]. These materials do not represent the composition of the local bedrock and are often better suited for large river basins such as that of the Amazon and the Congo [56], the Patagonian rivers, the Sebou river in Morocco, or the Garonne river in France [53]. These reference materials will have to be handled with caution [47,57]. To minimize these risks of errors, some authors use data from the local source of rock or uncontaminated fluvial sediments where they exist [40,44,46,58,59] because they reflect the geochemical background of the environment. Some others compare the data from the local source rock and that from the earth's crust for calculating EF [41,45,50]. The use of local material as a reference makes it possible to set the limit of the natural geochemical background at 1.5 [50,54] rather than 2, which is often used when the reference material used is not local [42,45,60]. It defines the degree of enrichment as follows, according to the EF value: EF < 2 no enrichment, 2 < EF < 5 moderate, 5 < EF < 20 significant, 20 < EF < 40 very high and EF > 40 extreme.

The reference materials used in this study are local references determined by Reference [59]. They correspond to two stream bottom sediments sampled in the Haho catchment, located close to our study area in the coastal sedimentary basin of Togo. One is collected in the upper part of the Haho basin (station H3 on the upper Haho river, see Reference [59]) and the other in a small creek, the Medje, draining a pristine small forested catchment located in the upper part of the Haho river basin (see Reference [59]).

The second important parameter in the EF calculation is the choice of the reference element (or normalizer). Its choice is not universal and depends on the lithological and physico-chemical characteristics of the study area [55]. A through literature, several elements such as Al [26,41,44,47,61,62], Cs [50,54], Fe [4,48,63], Li [64], Sc [45,58], and the organic matter [65] were used as references to calculate EF. Remaining in the same concept, other authors have also made comparative analyses using at least two reference elements [8,42,46], leading to various results depending on the benchmark used.

As reference elements for this study, aluminum (Al), cesium (Cs), iron (Fe), and titanium (Ti), were selected. The choice of Al was motivated by the fact that it is a natural and conservative element and a major constituent of clay minerals. It is frequently used during coastal and estuarine studies, even if it can introduce spurious correlations between variables, as discussed by Reference [62]. In a comparative study, Reference [66] revealed that Cs, structurally combined with clay minerals, could be used for estuarine and coastal sediments. It is also a good indicator for fine sediment fractions (<20 μm), which are phases strongly associated with TE. Fe was retained for its lithospheric origin [67] and for the fact that its distribution in clay fractions is not linked to other TE [68]. Finally, Ti was chosen for its natural origin and because it is supposed to be conservative during the aging process [69].

2.4.2. Sediment Quality Assessment Based on Sediment Quality Guidelines (SQG)

In order to facilitate the assessment of sediment quality, different sediment quality guidelines (SQGs) have been developed, based on the chemistry of the sediment and the corresponding adverse biological effects [70–74]. Many studies are based on these recommendations to assess the quality of sediments, whether freshwater, estuarine, or marine sediments, in order to propose suitable measures for the ecological management of aquatic environments [3,6,75–82]. Within the framework of our study, seven trace elements (As, Cd, Cr, Cu, Ni, Pb, and Zn) for which the SQG data exist in the literature were compared with their total concentrations in the Lomé lagoon sediments(See Supplementary Materials).

2.4.3. Calculation of Ce and Eu Anomalies and $(La/Yb)_N$ Ratio in Rare Earth Elements (REE)

For the study of rare earth elements (REE), normalization was carried out with the PAAS [51], which is often used to study REE in sediments of rivers or lagoons [26,40,42,50,53,83]. Ce and Eu anomalies, respectively, Ce* and Eu* are calculated as follows.

$$R = \text{concentration of the normalizing element}$$

$$Ce^* = Ce_N/(2La_N + Nd_N) + 2/3$$

$$Eu^* = Eu_N/(Sm_N \times Gd_N)^{1/2}$$
$$\text{with N = normalized.}$$

When the value of Ce* or Eu* is >1 or <1, the anomaly is said to be positive or negative, respectively.

To evaluate the fractionation between the light rare earth elements (LREE) and the heavy rare earth elements (HREE), one classically calculates the ratio $(La/Yb)_N$. When the value of the ratio is >1 or <1, there is, respectively, an enrichment in LREE compared to HREE or a depletion of LREE compared to HREE.

*2.5. Statistical Analyses and Data Treatment*

Data entry was possible using Excel software and data processing was carried out using STATISTICA 6.1 software (StatSoft Europe, Hamburg, Germany). The latter made it possible to establish the degrees of correlation between the variables and to compare the average values obtained during the dry and the wet seasons, using the Student's t-test. The geographic map is produced using QGIS software version 2.18 (Free and Open Source Geographic Information Suystem, QGIS Development Team, https://qgis.org/).

**3. Results and Discussion**

*3.1. Granulometry and Micro-Granulometry of the Lomé Lagoon Sediments*

The distribution of the different fractions shows that the sediments of the Lomé lagoon evolve from a coarse pole (sand) to a fine pole (clay and silt) represented, respectively, by the sites S7 of East Lake and S6 of Lake Bè (Figure 2a). These sediments are mainly composed of fine fractions (<63 μm), which represent 45% to 95% of the total sediment, respectively. The sediments of Lake Bè (S4–S6) are the finest, while those of East Lake (S7–S9) are the coarsest. The sediments of West Lake (S10–S12) are between the two.

As seen in Figure 2b, the finest fractions (<63 μm) are dominated by the fine silts (2–20 μm), which vary little from 52% (S4) to 66% (S5). The distribution patterns of the fine fractions evolve from clays (<2 μm), which represent 6% (S9) to 36% (S4), to coarse silts (20–63 μm), which are between 7% (S6) and 36% (S9). Among the finest sediments, the stations of Lake Bè (S4–S6) present the highest percentages (22% to 36%) of clay fractions.

Finally, the sediments of the Lomé lagoon are mostly composed of fine fractions, particularly fine silts.This reflects a mild hydrological regime of the water discharged into the Lomé lagoon. However,

these results differ from those found by Reference [26] for which the coarse sand fraction is the most abundant (55.3% to 60.1%). This difference can only be explained by the grain size analysis technique used in this study (laser microgranulometry), which is more efficient than the classical sieving used by Reference [26], which overestimates the coarse fractions.

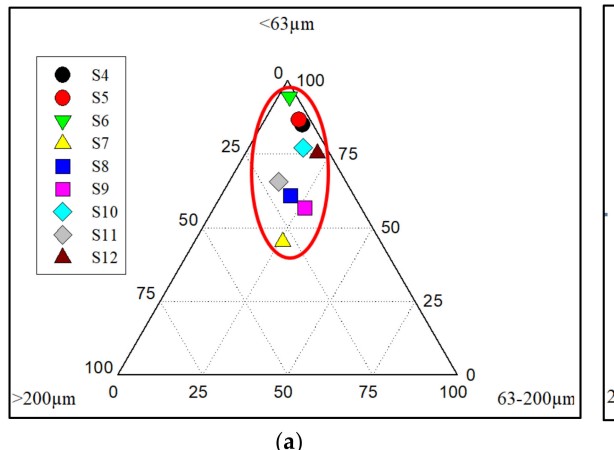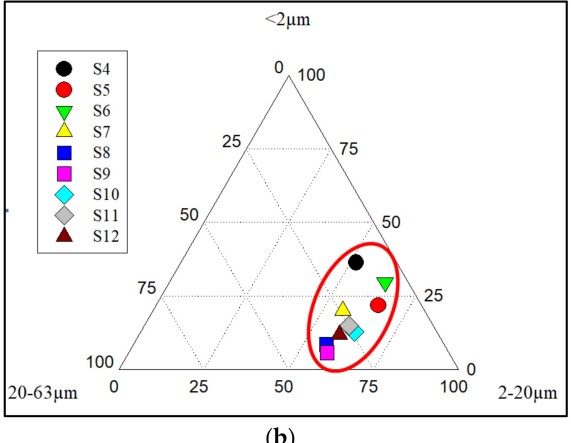

**Figure 2.** Ternary textural diagrams of bottom sediments of the Lomé lagoon: (**a**) for the total fractions (<2 mm), (**b**) for the fine fractions (<63 μm). S4 to S12 are the different sampling stations (see locations in Figure 1).

*3.2. Major Element Composition of the Lomé Lagoon Sediments*

The ternary diagrams for major element percentages (see Table S1 in the Supplementary Materials (SM) for basic data) show that the chemical composition of Lomé lagoon sediments (fine fractions <63 μm) varies little whatever the Lake, the station, or the season.

As in most of the river and lake sediments in the world, the silica (more than 50%) and the iron and aluminum oxides (10%–45%) are the major constituents of the Lomé lagoon sediments. The sum of major cations ($CaO + MgO + Na_2O + K_2O$) represents only 5% to 25%. This sediment composition is characteristic of the minerals constituting the lateritic soils of the study area (quartz, kaolinite, gibsite, goethite, hematite, etc.). The seasonal variations are not significant except for $SiO_2$ contents, which are, on average, significantly higher ($p$ value < 3%) during the dry season (54% to 67%, Figure 3a) than during the wet season (48% to 61%) (Figure 3b), particularly for the sites S7–S9 of East Lake.

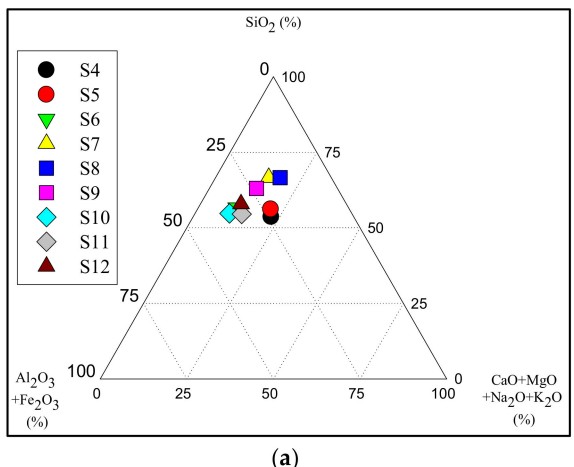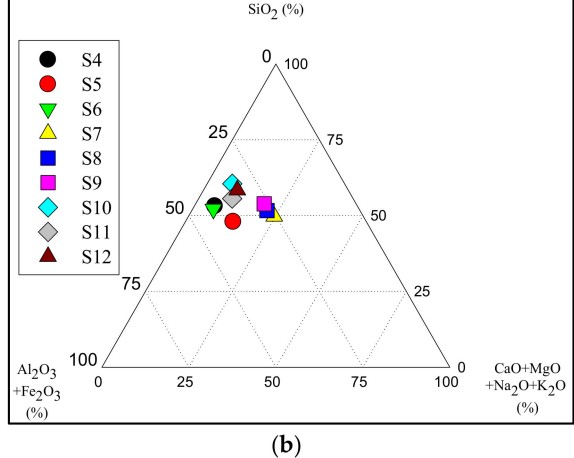

**Figure 3.** Chemical composition of the fine fractions (<63 μm): ternary diagrams of the major element percentages (% of the sum $SiO_2 + Al_2O_3 + Fe_2O_3 + CaO + MgO + Na_2O + K_2O$) in the bottom sediments collected in the dry season (**a**) and in the wet season (**b**). See Table S1 in Supplementary Materials for the basic data with Loss-on-ignition (LOI), MnO, $TiO_2$, and $P_2O_5$.

### 3.3. Rare Earth Element (REE) Content and Seasonal Variability

The average REE concentrations in the Lomé lagoon fine sediments (fractions <63 μm) are reported in Table 2 for both seasons and in Table S2 in Supplementary Materials for all the sites. The REE concentrations in the local reference material Sed_R are one to two times higher than those of the REE in the sediments of the Lomé lagoon unlike those of the PAAS. The REE concentrations in the Lomé lagoon sediments are, on average, higher during the dry season than during the wet one, particularly for light rare earth elements (LREE from La to Sm) for which the differences between both seasons are significant (*p* values < 5%). The LREE concentrations are also, on average, higher than those of the heavy rare earth elements (HREE from Eu to Lu), particularly during the dry season.

However, if we consider the different sites (S4 to S12), one can observe some differences in the global distributions of LREE versus HREE and of a dry season versus wet season. Then, in order to highlight these differences, it is better to normalize the REE concentrations in each site sediment (Figure 4) with the PAAS [51].

**Table 2.** Rare earth elements (REE) average concentrations in the fine sediments (<63 μm) of the Lomé lagoon during the dry and the wet seasons, compared with the contents in PAAS and in the local reference material, Sed_R. See also the concentrations for each site in Table S2 (Supplementary Materials).

| REE ($\mu g \cdot g^{-1}$) | Sediments of the Lomé Lagoon (n = 9) | | | | | | | | PAAS | Sed_R |
| | Dry Season (DS) | | | | Wet Season (WS) | | | | | |
| | Min | Max | Mean | Med | Min | Max | Mean | Med | | |
|---|---|---|---|---|---|---|---|---|---|---|
| La | 29.8 | 61.8 | 46.9 | 47.2 | 21.6 | 49.0 | 35.1 | 35.3 | 38.2 | 63.7 |
| Ce | 65.5 | 127.8 | 100.4 | 103.6 | 48.6 | 105.4 | 78.3 | 78.8 | 79.6 | 131.3 |
| Pr | 6.9 | 13.0 | 10.2 | 10.3 | 5.0 | 10.8 | 8.0 | 8.1 | 8.8 | 15.0 |
| Nd | 26.8 | 45.1 | 37.2 | 38.0 | 19.0 | 39.3 | 29.8 | 29.6 | 33.9 | 56.1 |
| Sm | 5.2 | 8.5 | 7.1 | 7.3 | 3.8 | 7.8 | 5.9 | 5.9 | 5.5 | 10.9 |
| Eu | 1.1 | 1.9 | 1.5 | 1.5 | 0.7 | 1.6 | 1.2 | 1.2 | 1.1 | 2.2 |
| Gd | 4.1 | 6.6 | 5.5 | 5.7 | 3.2 | 5.9 | 4.6 | 4.5 | 4.7 | 8.9 |
| Tb | 0.6 | 1.0 | 0.8 | 0.8 | 0.5 | 0.9 | 0.7 | 0.7 | 0.8 | 1.3 |
| Dy | 3.6 | 5.8 | 4.7 | 5.0 | 3.3 | 5.2 | 4.2 | 4.1 | 4.7 | 8.0 |
| Ho | 0.7 | 1.2 | 0.9 | 1.0 | 0.7 | 1.1 | 0.9 | 0.9 | 1.0 | 1.7 |
| Er | 2.0 | 3.1 | 2.4 | 2.4 | 2.1 | 3.0 | 2.4 | 2.5 | 2.8 | 4.5 |
| Tm | 0.3 | 0.4 | 0.3 | 0.3 | 0.3 | 0.4 | 0.4 | 0.4 | 0.4 | 0.7 |
| Yb | 1.9 | 3.1 | 2.4 | 2.2 | 2.3 | 3.2 | 2.6 | 2.5 | 2.8 | 4.6 |
| Lu | 0.3 | 0.5 | 0.4 | 0.3 | 0.4 | 0.6 | 0.4 | 0.4 | 0.4 | 0.7 |

Then PAAS normalization (Figure 4) of the Lomé lagoon sediments shows that all the sites present the same distribution patterns for each season but some differences can be noted between the dry and the wet seasons.

For the REE in the Lomé lagoon sediments, which are more concentrated during the dry period than during the humid one, the PAAS normalized distributions show the same patterns. Moreover, the REE concentrations in the Lomé lagoon sediments are higher than those in the PAAS, except for the sediments of the East Lake (S7–S9), which are less concentrated than the PAAS. S7–S9 sediments are enriched in silt fractions and in silica and cations when compared to the other sediments (see Figures 2 and 3, respectively). As already shown by different authors [84–86], REE have a greater affinity for clay fractions than for silts and sands. Coarse-grained sediments have little potential to immobilize REEs, while clay minerals are rich sources of REEs because of their ability to adsorb REEs on their surfaces and to incorporate them into their crystal structures. It is also important to notice that the REE fractionation between LREE (enrichment) and HREE (depletion) with regard to the PAAS is much more important during the dry period than during the humid one, except for two sediments of the East Lake (S7 and S8). These two sediments are enriched in HREE during the wet period likely because of their particle size (coarser) and of their chemical composition (enriched in silica and cations) as

mentioned above. Two classical positive anomalies appear for most of the sediments including one for Ce and the other for Eu.

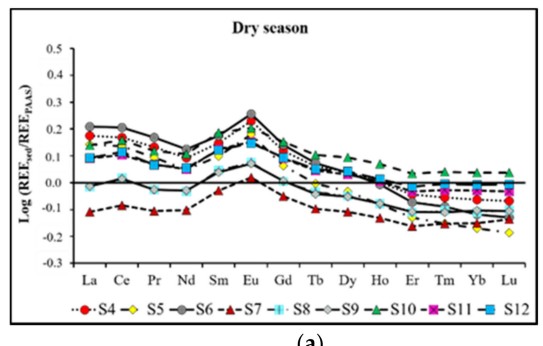
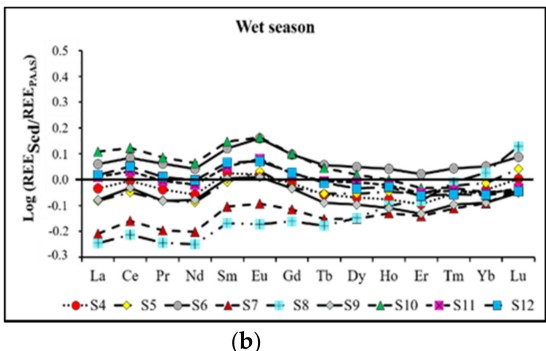

|  |  |
|:-:|:-:|
| (**a**) | (**b**) |

**Figure 4.** Distribution patterns of REE concentrations normalized to the PAAS concentrations in the fine sediments (<63 μm) of the different Lomé lagoon sites (S4 to S12) during the dry (**a**) and the wet (**b**) seasons. See the concentrations for each site in Table S2 (Supplementary Materials).

The Ce anomalies (Ce*) are less noticeable (no significant difference at 1% level) and less variable between seasons than Eu anomalies (Eu*), which are significantly different (*p* values < 1%). These two positive anomalies are linked to a change in oxidation state. In fact, unlike the other REEs, Ce and Eu in certain redox conditions have the capacity to change their state (from $Ce^{3+}$ to $Ce^{4+}$ and from $Eu^{3+}$ to $Eu^{2+}$), resulting in a modification of their geochemical behavior with consequent positive or negative anomalies [87,88]. The positive Ce anomaly would be due to the abundance of iron and manganese oxides, which are privileged sites for the adsorption of Ce in the sediments of the Lomé lagoon as underlined by Reference [89]. However, instead of a positive correlation, the results showed an earlier decrease in the Ce anomaly when the percent of Fe increases (Y = −235.48X + 248.28, $R^2$ = 0.7837). Reference [40] made the same observation in the sediments of the Ibrahim river. Then, the total Fe content is likely more due to Fe in clay minerals rather than in Fe-oxides. The anomaly in Eu would likely be linked to phases of substitution of elements like Ca, Sr, Na, and Zr, which are abundant in the sediments of the lagoon. These results are similar to those found in the Piracicaba basin in Brazil [90] and in the sediments of the Milo river at Kankan [42], which drain a silicate substratum with lateritic red soils.

The values of the normalized ratio (La/Yb) (Table 3) show that there is a fractionation between the REEs with an enrichment of the LREE compared to the HREE. This enrichment is more marked in the dry season than in the rainy season (significant difference at a 1% level), as previously shown from the concentration distributions. In the dry season, the absence of rain and water inflow facilitate the deposition of the finest particles at the bottom of the lagoon and, thus, promote the adsorption of REEs, in particular LREEs with a lower ionic radius than HREEs, which are difficult to resuspend during the rainy season. Then, the REE fractionation between LREE and HREE is partly controlled by the particle size and the Fe+Al concentrations in the clay mineralogical fractions. This is supported by the very good relationships between the La/Yb normalized ratios and the particle size (Figure 5, left) on one hand and the percentage of aluminum and iron oxides ($Al_2O_3$ + $Fe_2O_3$) on the other hand (Figure 5, right). Consequently, there is no significant correlation between $(La/Yb)_N$ and the percentages of silica ($SiO_2$) or the percentage of cations ($CaO$ + $MgO$ + $Na_2O$ + $K_2O$). It is also important to underline that there are significant negative relationships between Zr concentrations and $(La/Yb)_N$ for the dry ($R^2$ = 0.796) and for the wet ($R^2$ = 0.867) seasons because coarse fractions are enriched in zircons.

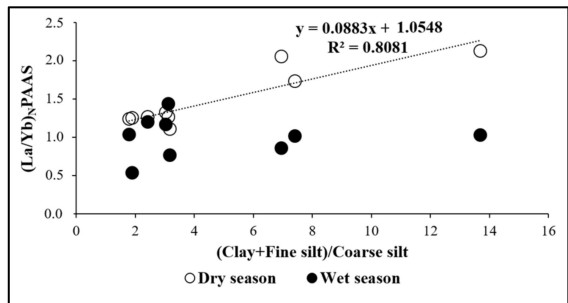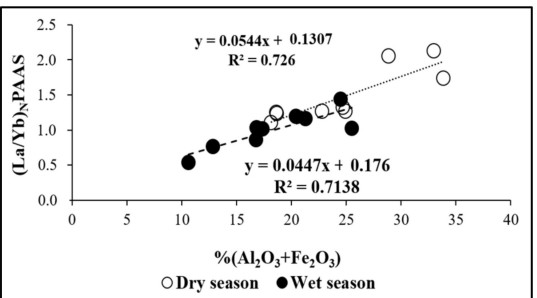

**Figure 5.** Relationships between the average $(La/Yb)_N$ ratios and the abundance of the finest particles on one hand (**left**) and the average percentage of $(Al_2O_3 + Fe_2O_3)$ on the other hand (**right**) for the Lomé lagoon fine sediments (<63 µm) during the dry and wet seasons.

**Table 3.** Average Ce and Eu anomalies (Ce* and Eu*, respectively) and La/Yb normalized ratios $(La/Yb)_N$ in the Lomé lagoon fine sediments (<63 µm) for dry and wet seasons. Eu* and $(La/Yb)_N$ average values are statiscally different between the two seasons (*p*-values < 1%).

| PAAS Normalization | Season | Min | Max | Mean |
|---|---|---|---|---|
| Eu* | Dry season | 1.08 | 1.25 | 1.16 |
| | Wet season | 0.98 | 1.13 | 1.07 |
| Ce* | Dry season | 1.01 | 1.03 | 1.02 |
| | Wet season | 1.02 | 1.04 | 1.03 |
| $(La/Yb)_N$ | Dry season | 1.10 | 2.12 | 1.48 |
| | Wet season | 0.53 | 1.44 | 1.00 |

*3.4. Trace Element (TE) Composition of the Lomé Lagoon Sediments*

3.4.1. Seasonal Variability

Tables S3 and S4 as well as Figure 6 reveal that the TE concentrations in the fine fractions (<63 µm) vary little, according to the seasons. Nevertheless, Cd, Cu, Hf, Pb, Sb, Sn, U, Zn, and Zr have higher concentrations during the wet season, but only Cd, Hf, Sn, U, and Zr present significant differences (*p*-values < 5%) between the two seasons. On the contrary, Bi, Ga, Ge, Ni, Rb, and Sc present significant enrichment (*p* values < 5%) during the dry season. Nevertheless, the two seasons present the same distribution pattern. However, the concentrations are greatly variable from one element to another, going $10^{-1}$ for trace elements like Bi to $10^3$ µg·g$^{-1}$ for minor elements like Zn and Zr. In the rainy season, concentrations range from values close to $3.8 \times 10^{-1}$ µg·g$^{-1}$ for Cd to $2.4 \times 10^3$ µg·g$^{-1}$ for Zr and, in the dry season, values range from $10^{-1}$ µg·g$^{-1}$ for Cd to $9.50 \times 10^2$ µg·g$^{-1}$ for Zn. The most abundant TE are Zn > Zr > Sr > Ba > Cr > V > Pb > Cu in the dry season and Zr > Zn > Sr > Ba > Pb > Cu > Cr > V in the rainy season with orders of magnitude between $10^2$ to $10^3$ µg·g$^{-1}$ regardless of the season. The TE concentrations measured in our sediment samples are higher than the concentrations found in the past in the lagoon sediments by Reference [26] for elements like Ni, Co, Cu, Mo, Cr, and Pb and by References [24,25] for Cd, Cu, Pb, and Zn.

As seen in Figure 6, TE concentrations in the Lomé lagoon sediments present the same distribution patterns than those of the PAAS, UCC, and the local materials Sed_R and H3, except for some elements like As, Cu, Mo, Pb, Sb, Sn, and Zn, which are more abundant in the Lomé lagoon sediments.

3.4.2. Enrichment Factors (EF)

Choice of Reference Material

The distribution patterns of the annual average concentrations of TE in the Lomé Lagoon sediments are normalized in Figure 7 by the different reference materials used in this study (PAAS, UCC, Sed_R, and H3). Some TE give, more or less, the same positive (As, Bi, Cd, Cu, Mo, Pb, Sb, Sn, Ta, U,

Zn) or negative (Ba, Be, Cs, Ge, Nb, Th) significant anomalies whatever the reference material used. On the contrary, for some other elements like Co, Cr, Cu, Ga, Hf, Ni, Rb, Sr, V, Y, and Zr, the values calculated using the different reference materials are different. Among these TE, Zr is mainly of natural origin. As already shown by References [42,59], Zr is enriched in the sediments of this African region mainly because of the erosion of lateritic soils, which cover most of the river basins. The lateritization of the soil profile by weathering processes leads to the accumulation of weathering-resistant minerals enriched in Zr like zircon [91]. Then, when normalizing with PAAS or UCC, the Zr anomaly appears to be positive whereas, when using the local reference materials, particularly the sediments of the Medje creek (Sed_R), there is no anomaly.

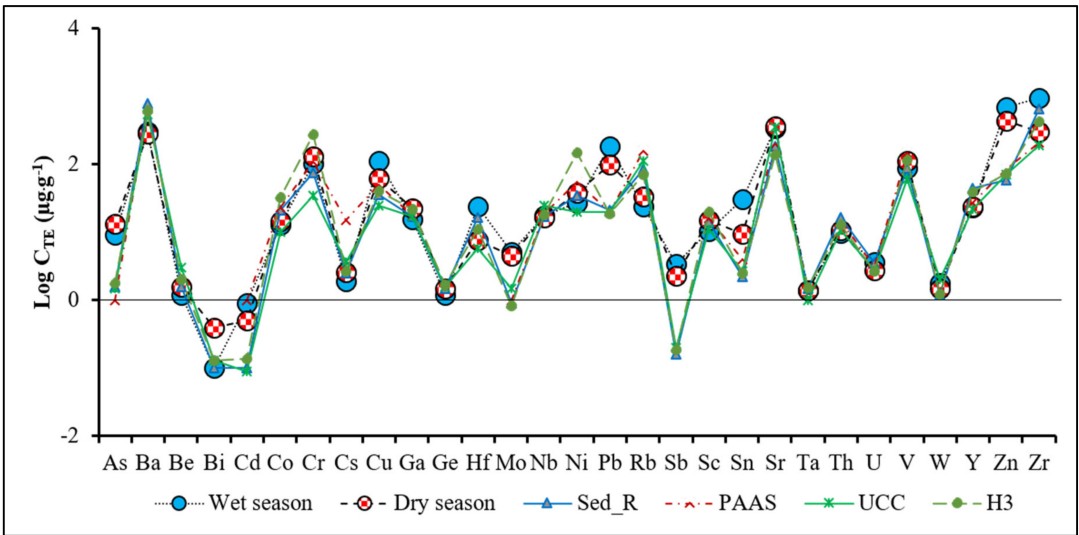

**Figure 6.** Average seasonal patterns of trace element (TE) concentrations in the Lomé lagoon sediments (means of nine site samples) during the dry and the wet seasons when compared with the PAAS, UCC, and the local reference materials, Sed_R and H3. See Table S3 for the average TE concentrations and Table S4 for TE concentrations for each site and each season.

Consequently, in the context of this study, for calculating the EF, the reference material Sed_R will be used to take the natural regional background into account.

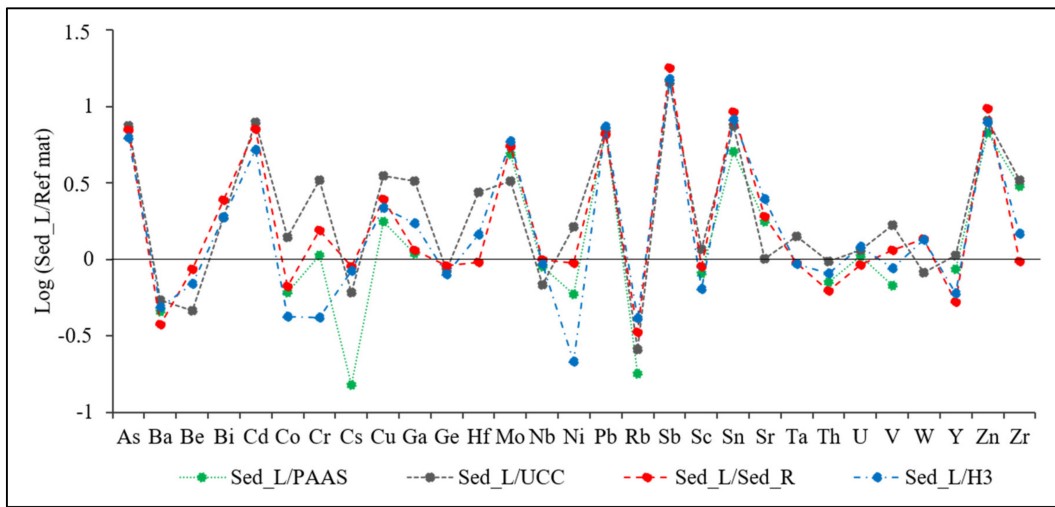

**Figure 7.** Normalization of average annual TE concentrations in the Lomé lagoon sediments (Sed_L) using different reference materials (PAAS, UCC, Sed_R, and H3).

Choice of the Normalizing Element

As already mentioned in Section 2.4.1, four TE (Al, Cs, Fe, and Ti) have been retained for normalization because these TE have very little contamination and they are mainly present in the residual fractions, as shown in Section 3.4.3. A single EDTA extraction, which allows us to remove all the labile fractions, reveals that Al, Cs, Fe, and Ti are mainly associated with the residual fractions that represent 97% to 99% of their total concentrations.

A comparison of the EFs calculated using the four normalizing elements (Al, Cs, Fe, and Ti) is presented in Figure 8. For each TE, the EF values are variable according to the choice of the normalizer as already pointed out by several authors [41,42,45,50,54], but all the normalizers exhibit the same significantly enriched TE (As, Cd, Mo, Pb, Sb, Sn, and Zn) and the ranges of values are consistent. The normalizations with Al, Cs, and Ti exhibit close EF values when compared with Fe. Moreover, Fe presents the worst correlations with the different TE (see Table 4). Then, in the context of this study, we can retain the EF values calculated with Al, Cs, and Ti as normalizers because the values are very close and they present the best correlations with the other TEs.

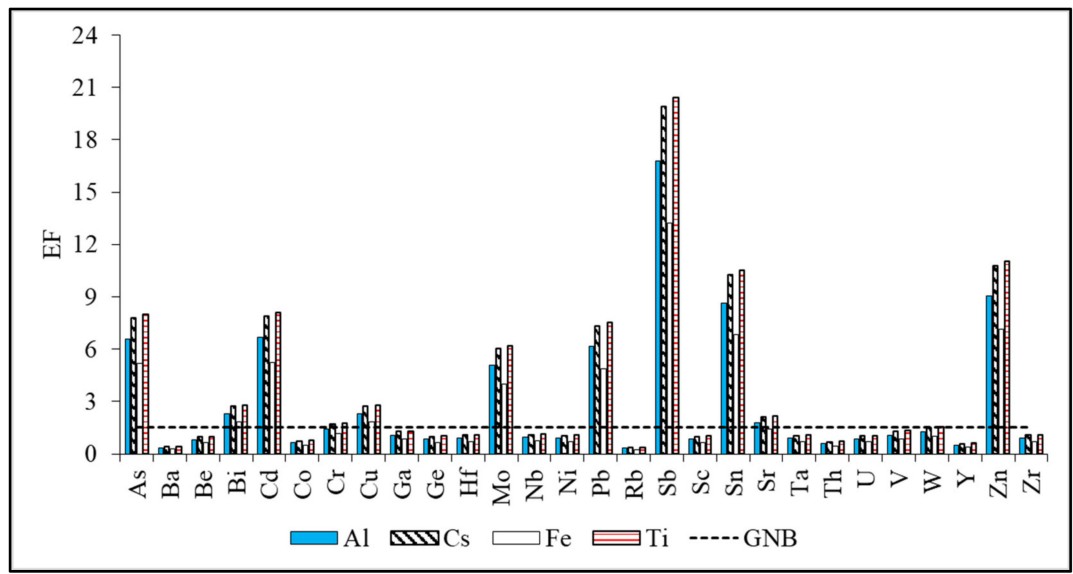

**Figure 8.** Comparison of EF calculated for each TE using different normalizers (Al, Cs, Fe, and Ti). GNB, geochemical natural background limit.

**Table 4.** Correlation matrix (r, Pearson correlation coefficient) between the different TE and the normalizing elements, Al, Fe, Cs, and Ti. The bold numbers have *p*-values at a 5% significance level.

|  |  | As | Cd | Co | Cr | Cu | Mo | Ni | Pb | Sb | Sn | Th | U | Zn |
|---|---|---|---|---|---|---|---|---|---|---|---|---|---|---|
| Al | Wet season | 0.54 | −0.26 | −0.03 | **0.83** | −0.55 | −0.31 | 0.18 | −0.46 | −0.40 | **−0.82** | 0.58 | −0.16 | **−0.75** |
|  | Dry season | −0.04 | 0.17 | 0.09 | 0.60 | 0.13 | **−0.67** | 0.66 | −0.11 | 0.05 | 0.13 | **0.76** | **−0.71** | 0.27 |
| Fe | Wet season | 0.40 | −0.15 | −0.24 | 0.36 | −0.43 | −0.09 | −0.28 | −0.55 | −0.24 | −0.52 | 0.26 | 0.18 | −0.66 |
|  | Dry season | 0.10 | −0.02 | −0.11 | 0.50 | 0.06 | −0.35 | 0.41 | −0.03 | −0.05 | 0.03 | 0.31 | −0.40 | 0.03 |
| Cs | Wet season | −0.30 | **−0.86** | **0.88** | 0.54 | −0.65 | 0.15 | **0.94** | −0.49 | −0.73 | −0.64 | **0.74** | **−0.87** | −0.34 |
|  | Dry season | −0.33 | 0.53 | 0.59 | 0.31 | 0.41 | **−0.68** | **0.76** | 0.16 | 0.33 | 0.48 | **0.96** | −0.52 | 0.64 |
| Ti | Wet season | −0.51 | −0.59 | **0.85** | 0.26 | −0.21 | 0.15 | **0.83** | −0.07 | −0.31 | −0.27 | **0.68** | **−0.76** | 0.14 |
|  | Dry season | −0.48 | 0.30 | 0.20 | −0.54 | 0.33 | 0.12 | −0.38 | 0.44 | 0.37 | 0.28 | −0.04 | 0.61 | 0.17 |

Enrichment Factors of Trace Elements

As seen in Figure 8, the most enriched TE are, on average, As, Cd, Mo, Pb, Sb, Sn, and Zn, which present significant enrichment ($5 < EF < 20$) and, to a lesser extent, Bi, Cr, Cu, and Sr, which are moderately enriched ($1.5 < EF < 5$).

Figure 9 presents only EFs of the most enriched sites at each Lake (S4 for Lake Bè, S8 for East Lake, and S11 for West Lake). The enrichment factors calculated overall show an enrichment of TE in the wet season to be greater than the enrichment in the dry season, except for some stations and some TEs (Cd, Sb, and Zn in S4, Cd, Cu, Pb, Sn, and Zn in S8). The TEs, which present a significant enrichment (As, Cd, Mo, Pb, Sb, Sn, and Zn), are always the same regardless of the lake and the station. Moreover, the level of enrichment is much higher (significant to very high) for S8 than for the other stations, particularly during the dry season.

In the wet season, the S10 site in West Lake records 28% of the enriched TE ($EF > 1.5$) against 83% of the TE enriched for the S8 site in the East Lake. The sites S6 at Lake Bè and S7 at East Lake record, respectively, the minimum (21%) and the maximum (41%) of the enriched TE during the dry season.

Whatever the season, the degrees of enrichment are moderate to significant for all the sites of Lake Bè and West Lake.

For the sites of Lake Bè (S4–S6) in the dry season, the TE such as As, Mo, Pb, Sn, and Zn show moderate degrees of enrichment while Cd and Sb are significantly enriched. In the wet season, Bi, Cd, Cr, Ga, Mo, Pb, Sn, V, and Zn are moderately enriched while As and Sb are significantly enriched.

For the West Lake sites (S10–S12) in the dry season, TE show moderate (As, Cd, Mo, and Pb) to significant (Sb, Sn, and Zn) enrichment degrees. In the wet season, only Cu and Mo are moderately enriched while As, Bi, Cd, Pb, Sb, Sn, and Zn are significantly enriched.

For the East Lake sites (S7–S9) during the dry season, Hf and Sr are moderately enriched while As, Cd, Cu, Mo, Pb, and Zn are significantly enriched. Sb and Sn show very high enrichments. During the wet season, only Cu is moderately enriched. As, Bi, Cd, Mo, Pb, Sn, Sr, and Zn are significantly enriched and Sb exhibits a very high enrichment.

The EFs of the other TE (Ba, Rb, and Y for the three lakes, plus Co, Cs, Ge, Hf, Nb, Ta, Th, U, W, and Zr for Lake Bè and West Lake, plus Be, Ga, Ni, Sc, Sr, and V for West Lake) are below the regional geochemical background level. The elements with significant EF values are generally recognized for their anthropogenic origin. In the context of this study, these elements are essentially linked to urban pollution. As pointed out by References [24–26], this urban pollution consists of (i)- all the wastewaters from various domestic and collective uses (households, health centers, schools, markets, etc.), which are discharged into the lagoon without any treatment, but also of (ii)- the traffic of cars and motorcycles (brake wear and past leaded gasoline) [92,93], and of (iii)- the atmospheric fallout from the combustion (fuel, petrol, incineration of household waste) and the deposit of fine particles in the lagoon. Agricultural pollution is lower and would be due to the use of chemical fertilizers, particularly for Cd, as already shown by Reference [50] and unauthorized phytosanitary products. Furthermore, EF calculated revealed that Ba, Cr, and Zr whose total concentrations range between $10^2$ to $10^3$ $\mu g \cdot g^{-1}$ (see Tables S3 and S4 in Supplementary Materials) and are not enriched and lower than the limit of the regional natural geochemical background ($EF < 1.5$). Then these elements can be considered as of a natural origin and linked to the geological substratum and to the lateritic soils formed through the chemical weathering processes. The geochemical variability of Ba is mainly caused by biogenic and textural factors rather than by anthropogenic activities [47].

The low EF values for Cr come from its lithological character, which can be influenced by anthropogenic contributions as underlined [45,54]. It is the same for Zr whose lithogenic origin has been demonstrated by References [40,42] and which is a component of zircon heavy residual minerals that accumulate in lateritic soils [5,91].

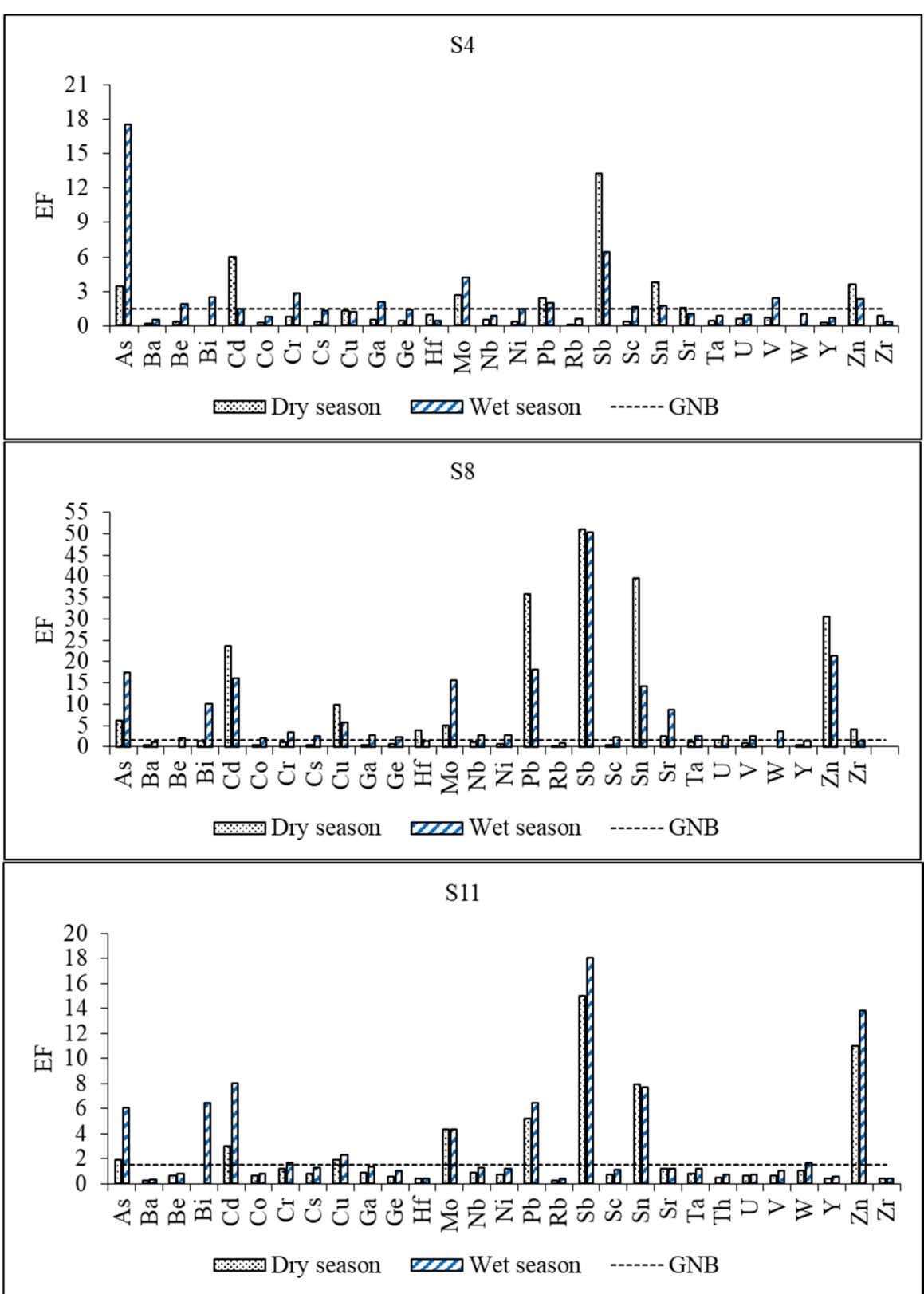

**Figure 9.** Distribution of TE enrichment factors (EF) for three sites (S4, S8, and S11) representative of each Lake (respectively Lake Bè, East Lake, and West Lake). EF are calculated with Al as a normalizing element (as an example, but Cs and Ti give, more or less, the same patterns) and using Sed_R as local reference material. The dashed lines represent the geochemical natural background (GNB). EF values < 1.5 are within the range of the natural regional geochemical background.

### 3.4.3. Distribution of TE between the Residual and Non-Residual (labile) Phases

The total extraction of the TEs and the calculation of the EFs made it possible to measure the total concentration of TEs and to determine their natural or anthropogenic origin without giving information on their bioavailability or lability in the Lomé lagoon sediments. Numerous studies [5,41,42,46,50,94,95] have shown the importance to undertake single or selective extractions because only the TEs in the labile phases represent a threat to living organisms and to aquatic ecosystems. The percentages of non-residual fractions (NRF) extracted using EDTA are reported in Table 5 for each element and for both seasons. The results show that the labile fractions represents less than 15% of the total concentrations except Co, Cu, Y (both seasons), and Sr (dry season) for which the NRF range between 15% to 25%, Pb, Zn (both seasons), and Sr (wet season) for which the NRF range between 25% to 40% and Cd (both seasons) between 47% to 55% regardless of the season (see Figure 10 left for Cd in the different sites). In addition to these elements, As, Bi, Ni, V, Mo, and Sc (both seasons) have NRF percentages between 5% to 15%. The anthropogenic contributions of all these TE will mainly tend to adsorb onto the sediments in the non-residual phases. Consequently, they can be released into the water column if there is any change in the physicochemical characteristics of the lagoon waters (pH, redox potential, salinity, dissolved organic carbon, and inorganic ligands). References [96–98] showed that TEs such as Pb, Cu, Zn, and Cd can be rapidly bio-available in the aquatic environment if the pH drops down to pH 4, 5, 5.5, and 6, respectively. TEs such as Ba, Cr, and Zr (see Figure 10 right for Zr in the different sites) whose natural origins have been demonstrated in the previous paragraph have percentages of labile fractions less than 1%. This confirms their natural origin in the Lomé lagoon sediments despite their high total concentration in the sediments (see Tables S3 and S4 in Supplementary Materials). Reference [40] has shown, in the sediments of the Ibrahim river in Lebanon, a good positive relationship between EF and labile fractions for elements of anthropogenic origin such as Cu, Co, Ni, Zn, and Pb and negative for Zr, which is of a natural origin. In the Milo River, sediments at Kankan (Guinea) [42] have also noticed good relationships between EF values and the percentages of labile fractions, which are positive for Co, Mn, Zn, and Sb and negative for Zr and Hf, which are of a natural origin. In the Lomé lagoon sediments, these relationships were also verified with negative correlation between the EF and percentage of NRF for Th ($Y = -6.56X + 3.99$, $R^2 = 0.55$), and Nb, which are of a natural origin and positive correlation for Co ($Y = 12.52 \ln(X) + 14.49$, $R^2 = 0.57$), which is of anthropogenic origin.

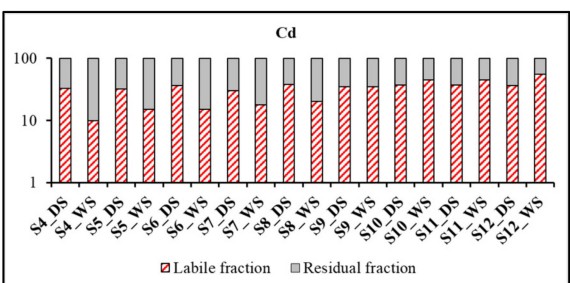 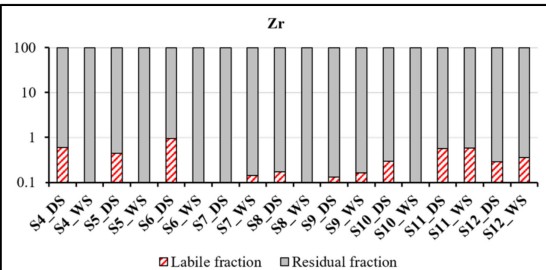

**Figure 10.** Distributions (percentage on a logarithmic scale) of non-residual (labile) and residual fractions for one TE of anthropogenic origin, Cd (**left**) and one TE of natural origin, Zr (**right**) in the different sediments of the Lomé lagoon (S4 to S12) for the dry (DS) and the wet (WS) seasons.

**Table 5.** Distribution of the NRF percentages for the different TE measured in the Lomé lagoon sediments.

| NRF (%) | <1% | 1–5% | 5–15% | 15–25% | 25–40% | 40–60% |
|---|---|---|---|---|---|---|
| Dry season | Al, Ba, Be, Cr, Cs, Ga, Ge, Nb, Rb, Sn, Ta, Th, Zr, Ti, W | Sb, Cs, Hf, U, Fe, | As, Bi, Ni, V, Mo, Sc | Co, Cu, Y, Sr | Pb, Zn | Cd |
| Wet season | Al, Be, Cr, Cs, Hf, Nb, Rb, Sn, Ta, Th, U, Zr, Ti, W | Sb, Cs, Ba, Ga, Fe, Ge | As, Bi, Ni, V, Mo, Sc | Co, Cu, Y | Pb, Zn, Sr | Cd |

### 3.4.4. Comparison with the Sediment Quality Guidelines (SQGs)

The total TE concentrations in the Lomé lagoon sediments compared to the SQGs [70–74], (Table S3) show that TE such as As, Cd, Cu, and Ni have total concentrations within the range of variation of the SQG concentrations, whereas Cr, Pb, and Zn total concentrations are higher than the SQG values.

According to Reference [99], PEL and ERM values can be used to estimate the probability of toxicity for sediment-dwelling organisms, using the PEL and ERM quotients, $Q_{PEL}$ and $Q_{ERM}$, respectively. $Q_{PEL}$ and $Q_{ERM}$ are the ratio between the ET concentration measured in the sediment and the corresponding PEL or ERM concentration. Then Reference [99] propose to use $Q_{PEL}$ and $Q_{ERM}$ values for ranking or prioritizing the sites with respect to sediment contamination and to the probability of toxicity for benthic organisms [100] (Table S5).

$Q_{PEL}$ and $Q_{ERM}$ values have been calculated for all the sites and for the two seasons when ERM and PEL concentrations were available (see Table S6 in Supplementary Materials). Then, with regard to As, Cd, and Cu concentrations, all the sites (except S8 for Cd and Cu) can be ranked as lowest or medium-low priorities according to the site, the season, or the TE. On the contrary, for the other TE, the different sites can be considered as medium-high (Cr and Ni) to highest (Pb and Zn) priorities, according to site/season/TE. The highest priorities concern mainly East Lake sites (particularly S7 for Zn, S8 and S9 for Pb and Zn) and West Lake sites (particularly Zn for all the sites). On the contrary, all the sites of Lake Bè (S4–S6) are not concerned by the highest priorities. Lake Bè can be ranked as medium-high priority with regard to Cr, Ni, Pb, and Zn, and medium-low to low priority for As, Cd, and Cu.

However, these guide values were calculated based on the total concentration of TEs. However, several studies today have shown that the total concentration of a trace element cannot characterize the toxicological and environmental risks of sediments because only bioavailable fractions can be toxic [50,101–105]. In the previous paragraph, the non-residual (labile) fractions, which have been extracted with EDTA and which represent the bioavailable fractions, are <1% of the total concentrations for Cr between 5% and 15% for As and Ni, 15–25% for Cu, 25–40% for Pb and Zn, and 40–60% for Cd. Then, if we consider mainly the NRF concentrations for each TE, all the sites we have studied in the Lomé lagoon will have concentrations below the SQGs. In particular, this is the case for Cr, As, Cu, and Ni, which have the lowest NRF percentages, with, consequently, less potential biological effects on the aquatic organisms. Such results have already been observed in the same kind of river basin covered by lateritic soil by Reference [42] in the Milo river sediments (Guinea) and by Reference [59] in the Zio and Haho river sediments (Togo).

## 4. Conclusions

The purpose of this study was to assess the spatio-temporal variations and distributions of major, trace, and rare earth element concentrations in the sediments of the Lomé lagoons located in an urban environment. The seasonal variability of major REE and trace element concentrations is not very marked. However, the study revealed a slight enrichment in silica during the dry season, particularly for the sediments of East Lake (sites S7–S9), which have more abundant sandy particles, and a larger fractionation between light REE and heavy REE (La/Yb > 1). The main phases of REE transport on sediments are mainly linked to aluminum and iron oxides and to the particle size distribution, especially to the finest fractions of the sediments. Ba, Cr, Cu, Pb, Sr, V, Zn, and Zr are the most abundant TE, but only Cu, Pb, and Zn have enrichment factors (EF) greater than the limit of the natural regional geochemical background. The comparison of EFs calculated with different normalizing elements (Al, Cs, Fe, and Ti) and the correlations established between the normalizing elements and the other TE showed that Al, Cs, and Ti give very close EF values and are the most suitable normalizers for the Lomé lagoon sediments. Thus, As, Bi, Cd, Cu, Mo, Pb, Sb, Sn, and Zn are the most enriched TEs, representing 31% of the studied TEs. This percentage varies according to the Lake/Site and to the season. It is going from 28% (S10 site in West Lake) to 83% (S8 site in the East Lake) in the wet season and from 21% (S6 site at Lake Bè) to 41% (S7 and S9 sites of the East Lake) during the dry season. The East Lake is

the most impacted by the urban activities of Lomé agglomeration. Ba, Be, Cr, Nb, Rb, Th, and Zr whose natural origins have been demonstrated by calculating the enrichment factor present percentages of non-residual fractions (NRF) less than 1% for both seasons. On the contrary, Cd, Pb, and Zn, which present significant enrichments, have NRF percentages in between 20% to 55% and are likely to be released into the water column in order to present a risk to the health of aquatic species.

A comparative analysis of TEs in relation to sediment quality guidelines allows us to conclude that TE such as As, Cd, Cu, and Ni have total concentrations within the range of variation of the SQG concentrations (particularly PEL and ERM), whereas Cr, Pb, and Zn total concentrations are higher than the SQG values. Moreover, PEL and ERM quotients allow us to determine the priority ranking for each site, showing that the priorities for almost all the sites are lowest to medium-low with regard to As, Cd, and Cu and medium-high (with regard to Cr and Ni) to highest (with regard to Pb and Zn), particularly for the East and West Lakes. The next step is to assess the impact of these TE on the fishes (ongoing study) and to carry out a characterization of the sediments of the Lomé lagoon in polycyclic aromatic hydrocarbons (PAH). This will make it possible to have the first PAH data and to propose solutions to strengthen the management of this lagoon ecosystem.

**Supplementary Materials:** The following are available online at http://www.mdpi.com/2073-4441/12/11/3026/s1, Table S1: Major element contents in the fine fractions (<63 μm) of the Lomé lagoon sediments during the wet and the dry seasons. S4–S6: Lake Bè sites, S7–S9: East Lake sites, S10–S12: West Lake sites. LOI: Loss-On-Ignition, which corresponds to the organic matter and the carbonates. Table S2: Rare earth element concentrations in the fine fractions (<63 μm) of the Lomé lagoon sediments during the wet and the dry seasons. S4–S6: Lake Bè sites. S7–S9: East Lake sites. S10–S12: West Lake sites. Table S3: Average trace element concentrations in the Lomé lagoon sediments (fraction < 63 μm) and TE concentrations of PAAS, UCC, two local materials (Sed_R and H3), and some Sediment Quality Guidelines (SQGs). See TE concentrations for each site in Table S4. Sediment Quality Guidelines for sediments (SQGs): TEL-Threshold Effect Level [73], ERL-Effects Range Low [70], LEL-Lowest Effect Level [72], MEL-Minimum Effect Level [71], PEL- Probable Effect Level [73], ERM-Effect Range Median [70], SEL-Severe Effect Level [72], TET-Toxic Effect Threshold [71], CB PEC-Consensus-Based for Probable Effect Concentrations [74]. Table S4: Trace element concentrations in the fine fractions (<63 μm) of the Lomé lagoon sediments during the wet and the dry seasons. S4–S6: Lake Bè sites. S7–S9: East Lake sites. S10–S12: West Lake sites. Table S5: Ranking of priority sites with respect to sediment contamination according to ERM and PEL quotients ($Q_{ERM}$ and $Q_{PEL}$) and to the probability (%) of toxicity for benthic organisms (after [99]). See $Q_{ERM}$ and $Q_{PEL}$ values calculated for Lomé lagoon sediments in Table S6. Table S6: PEL and ERM quotients, respectively, $Q_{PEL}$ and $Q_{ERM}$ calculated for all the sites (S4–S6 for Lake Bè, S7–S9 for East Lake and S10–S12 for West Lake) for the two seasons (WS, wet season and DS, dry season). Bold figures: highest priorities. *Bold italics*: medium-high priorities. Normal figures: medium-low priorities. *Italics*: low priorities (see Table S5 for the corresponding $Q_{PEL}$ and $Q_{ERM}$ limit values and for the toxicity probabilities).

**Author Contributions:** Conceptualization, T.E.-E.B. and J.-L.P. Methodology, T.E.-E.B. and J.-L.P. Validation, T.E.-E.B. and J.-L.P. Formal analysis, T.E.-E.B. Investigation, T.E.-E.B. and J.-L.P. Resources, T.E.-E.B., A.M.D.A., and K.O.-S. Data curation, T.E.-E.B. Writing-review and editing, T.E.-E.B. and J.-L.P. Visualization, T.E.-E.B. Supervision, J.-L.P., K.G., and S.J.-D. Project administration, J.-L.P. and K.G. Funding acquisition, J.-L.P. All authors have read and agreed to the published version of the manuscript.

**Funding:** This research was funded by the Cooperation and Cultural Action Service (SCAC) of the French Embassy in Togo for the doctoral fellowship of T.E.-E.B. in France and by the Laboratory of Functional Ecology and Environment on its contracts and its own CNRS resources.

**Acknowledgments:** The authors would like to thank the technicians and engineers of the Laboratory of Functional Ecology and Environment at Toulouse (Marie-José Tavella, Virginie Payre-Suc, Frédéric Julien, and David Baqué) for their welcome, advice, and technical assistance to the laboratory during the preparation and analysis of the samples. Thanks also to the SARM (CNRS analytical platform) in Nancy for the ICP-OES and ICP-MS analyses, and to A. Lanzanova & Camille Duquenoy (GET, France) for their technical support with the ICP-MS analyses (ICP-MS Platform, Midi-Pyrénées Observatory, Toulouse). We would like to thank colleagues from the GTVD laboratory (Ouro-Sama Kamilou, Solitoke D. Hodabalo, Sama Daouda) in Lomé for their help in the preparation/preliminary storage of the samples. Our thanks to the Cooperation and Cultural Action Service (SCAC) of the French Embassy in Togo who supported this study through the Campus France grant, which enabled Tchaa Esso-Essinam Badassan to carry out a three-month stay/year during three years in the Laboratory of Functional Ecology and Environment (Toulouse, France). This work has been achieved within the framework of an international jointly-awarded PhD program agreement between the University of Lomé (Togo) and Toulouse INP (France). The authors would like also to thank two anonymous reviewers for their very helpful and valuable comments, which allowed us to greatly improve the first version of the manuscript.

**Conflicts of Interest:** The authors declare no conflict of interest.

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
