# Peer review of "Geochemical Composition of the Lomé Lagoon Sediments, Togo: Seasonal and Spatial Variations of Major, Trace and Rare Earth Element Concentrations"

_water, doi:10.3390/w12113026_

Round 1

Reviewer 1 Report

This paper focuses on the geochemistry and sediment quality of Lome Lagoon in the southern part of Tongo. Total and EDTA-extractable contents of trace elements and rare-earth elements are discussed with respect to a spatial and seasonal basis, the enrichment or depletion in relation to the local background, as well as in relation to sediment quality guidelines.

After reviewing the manuscript, I recommend major revision prior to its publication. My major concerns are listed below, whereas further comments are annotated in the pdf file.

  1. I’m also a little bit confused with the term lagoon. I cannot see on the map the connection of the three systems to the sea. What is the salinity in the ‘lagoons’? Maybe some details about the hydrological regime could be added in the relative section to justify the term. Besides, the term lake is also used in the manuscript.
  2. The manuscript could be better structured to ensure a logical flow. For example, the concentrations of REE should be discussed first, and then their EFs. My suggestion is to re-order sections (after section 3.3) as: total concentrations of trace elements, total concentrations of REE, EDTA extraction, EFs, comparison to SQGs.

Furthermore, in section 3.3.3.2 “Enrichment factors...” I would suggest mentioning first which elements are enriched and which are depleted, and then moving to the details concerning the seasonal variation and figure 7.

  1. Several parts of the manuscript could be shortened, without losing the main meaning. Some examples are:
  • the argumentation in favour of the use of local background over world average values in section 2.4.1.
  • The conclusions, which are too long. Authors are encouraged to shorten the text to 1-2 paragraphs.
  • Figure 4 could be removed, as all relative info is presented in Table 2.
  • The first paragraph of section 3.3.1 (L. 275–281). Differences of orders of magnitude in the concentrations of the studied elements are expected as some of those are minor elements and some are trace elements.
  1. Most of the discussion focuses on the seasonal variability of the different variables. Statistical tests could provide solid evidence on whether seasonal differences exist or not. Furthermore, basis, some explanation on the observed differences between the two seasons should be given, as for example in the case of the differences on the extractability of TE (Y, Mo, V, etc) by EDTA.
  2. The choice of normalizing element should be based on concrete criteria and not on the resulting level of contamination. The correlation matrix of elements and potential normalizers is a sound approach; Caesium and Ti should be also included in Table 3. Authors may also consult the paper of Van der Weijden (https://doi.org/10.1016/S0025-3227(01)00297-3) for sound approaches.
  3. It is true that the potential mobility of trace elements is better assessed by their non-residual fractions, rather than their total concentrations. However, the proposed SQGs have been derived after copious and long-running research combining total concentrations of pollutants and the incidence of adverse effects on benthic biota. Therefore, I found the proposed approach of dividing the total concentrations by an extractability factor? (it is not straight-forward how the factors have been calculated) rather arbitrary. Furthermore, there is no evidence, or supporting literature, that the EDTA extractable fractions are linked or have been linked to “actual toxicity. Authors may consult the paper of Long and MacDonald (https://doi.org/10.1080/10807039891284956) on the recommended uses of SQGs.
  4. Grammar and language need to be improved.

Author Response

See the attached document in word format.

Reviewer 2 Report

Paper is generally fine. Methods well documented, data evaluation appropriate. Some problems with data presentation make it hard to evaluate the paper rigorously and will cause some problems for readers.

  1. You must show table of major elements as well as traces and REE
  2. I assume a data supplement will show sample results by site as well as by season. If not, such a table needs to be included in the main body of the paper.
  3. You show too many significant figures in Table 2. Be very careful to show the precision appropriate for each element at the level measured. Three significant figures is normally the max.
  4. You must include an analysis of variance to show whether differences among the sites and between the seasons are significant. You report wet and dry season Eu anomalies of 1.07 and 1.16. I doubt that these are statistically different. The Ce anomalies are clearly all identical.
  5. Heavy/light REE. Heavy REE are strongly enriched in zircons, which concentrate in the silt fraction. Your figure 10 (left) shows HREE enrichment in the coarse (silt) fraction, which I attribute to zircons. Check the correlation of Zr to HREE/LREE.
  6. You need to ask the questions Do all three sites receive sediment from the same source? Does that source change with season? That is, before any anthropogenic effects can be discerned, you have to eliminate natural variations. A traditional way to check for this is to look at the Nb/Y ratio and the Zr/Al ratio. Nb/Y does not vary for your wet v dry season average, but I cannot evaluate between site variation nor can I see the Zr/Al ratio. You need to do this preliminary screen to establish that there is not some basic geologic difference among your sites.
  7. You allude to enrichment of Sb and Sn at one site. These elements can come from brass corrosion. If so they should correlate to Cu and Zn. I bring this up because it can be real or can be an artefact of sampling materials.
  8. In your inventory of possible pollution sources consider runoff of metals from automobiles. Brake wear produces large amounts of metal in street dust. Also if Togo uses or recently used leaded gasoline, you will have a major impact from that source. There is a large literature on both of these sources. 

Author Response

(The authors gave the same response as above.)

Round 2

Reviewer 1 Report

The revised version has been improved and could be accepted for publication after minor revision with respect to the followings: 

(Lines correspond to the marked-changes version)

89–90: delete “with brackish waters (Total dissolved solids 89 between 2.1 and 2.8 g·L-1 according to the season)”; it is mentioned in L. 94–95.

92: Three, instead of 3.

113: Is it really plastic??? 

120: correct to “disaggregate”.

274: “cation oxides” is not a correct term. Please replace with “major elements”.

276–279: Seasonal variation should be discussed according to the results of statistical tests (either there is a seasonal variation or not); phrases like “very slight”, “slightly more enriched”, “slightly higher” are vague.

316–320: Sentence is too long.

337: Rephrase “more present”.

367: Again, a mention to the results of the t-test is missing.

416–426: This paragraph should be re-written. The level of contamination is a subject under research, and as such, it cannot be the criterion for choosing the most appropriate normalizer (L. 420–423).

Table 6 could be moved to Supplementary Material (No data are presented).

Long sentences should be avoided and in general, English could be further improved.

Author Response

See attached document in word format.
